

# Sequential adiabatic generation of chiral topological states

Xie Chen[1], Michael Hermele[2] and David T. Stephen[1,2]

**1** Institute for Quantum Information and Matter and Department of Physics,
California Institute of Technology, Pasadena, CA, 91125, USA
**2** Department of Physics and Center for Theory of Quantum Matter,
University of Colorado, Boulder, Colorado 80309, USA

## Abstract

In previous work, it was shown that non-trivial gapped states can be generated from a product state using a sequential quantum circuit. Explicit circuit constructions were given for a variety of gapped states at exactly solvable fixed points. In this paper, we show that a similar generation procedure can be established for chiral topological states as well, despite the fact that they lack a zero-correlation-length exactly solvable form. Instead of sequentially applying local unitary gates, we sequentially evolve the Hamiltonian by changing local terms in one subregion and then the next. The Hamiltonian remains gapped throughout the process, giving rise to an adiabatic evolution mapping the ground state from a product state to a chiral topological state. We demonstrate such a sequential adiabatic generation process for free fermion chiral states like the Chern Insulator and the $p + ip$ superconductor. Moreover, we show that coupling a quantum state to a discrete gauge group can be achieved through a sequential quantum circuit, thereby generating interacting chiral topological states from the free fermion ones.

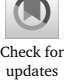

## Contents



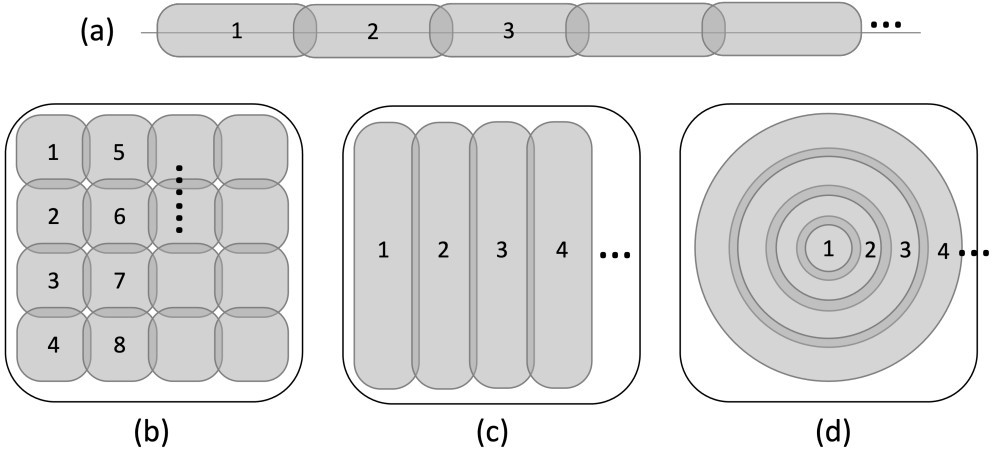

Figure 1: Sequential patterns in a sequential unitary transformation. A sequential unitary transformation can be a sequential quantum circuit or a sequential adiabatic evolution.

# 1 Introduction

In Ref. [1], it was proposed that to map from one gapped state[1] to another gapped state in a different phase, one needs a *sequential quantum circuit*. A sequential quantum circuit [2–8] is a quantum circuit generically of linear depth or higher, but where each layer acts on only a subregion in the whole system. The constrained structure ensures the entanglement area law is preserved and hence also the energy gap (if we start with a finite gap). On the other hand, the sequential quantum circuit does not necessarily preserve locality of operators, short-range correlation, or short-range entanglement, and is therefore potentially capable of mapping between states with different gapped orders.

In Ref. [1], this potential capability was shown to be achievable via explicit construction of sequential circuits that map from a product state to symmetry breaking states with long-range correlation, symmetry-protected topological states, topologically ordered states with fractional excitations, and fracton states. Such explicit constructions are possible because these phases have exactly solvable points with fixed point ground state wave functions of a simple structure.

Chiral phases are another important class of gapped phases, but they cannot be realized in exactly solvable models where the Hamiltonian is a sum of commuting terms [9–11]. Can ground states in chiral phases be generated in a similar way, maybe not using sequential circuits, but more generally with some form of sequential unitary transformation? In this paper, we answer this question in the affirmative by demonstrating how chiral states can be generated using *sequential adiabatic Hamiltonian evolution*. That is, we start from the gapped Hamiltonian of a product state, change the Hamiltonian in one sub-region, then change the Hamiltonian in the next sub-region, until the whole system is covered. We show that the Hamiltonian remains gapped during the whole process, so in each step the effect of Hamiltonian evolution on the ground state corresponds to a finite-time adiabatic evolution. The ground state is changed from a product state to a chiral state one sub-region at a time. The whole process takes a time that is proportional to the number of sub-regions, which is usually taken to scale linearly or higher in the linear size of the system. For the 2D chiral states discussed in this paper, we are going to use the sequential scheme as shown in Fig. 1(c) where

---

[1]A quantum state is called gapped if it is the ground state of a gapped local Hamiltonian.

in each step the Hamiltonian in a 1D slice is changed, and the slice being acted upon moves from left to right as the evolution proceeds.

In particular, we demonstrate the sequential adiabatic evolution for free fermion chiral states – the Chern insulator in Section 2 and the $p + ip$ superconductor in Section 3 – by showing through an explicit numerical calculation that the Hamiltonian remains gapped in the evolution process. We discuss a scheme based on the coupled wire construction [12,13], which gives physical intuition for the evolution of the state. The adiabatic process has some similarity to the one used in [14] for the entanglement renormalization transformation of chiral states. In the renormalization procedure of [14], as the length scale gets larger, the procedure involves the adiabatic evolution of nonlocal coupling terms. It is therefore a different setup than the one considered in this paper.

The results on free fermion chiral states pave the way for demonstrating sequential unitary generation of strongly interacting chiral states. We show in Section 4 that coupling a quantum state to a discrete gauge field can be realized with a sequential quantum circuit. Combining the sequential adiabatic evolution generating the free-fermion chiral state with a sequential quantum circuit that couples the system to a discrete gauge field, we arrive at a sequential unitary transformation that generates strongly interacting chiral states like the chiral Ising state, the chiral semion state, etc.

## 2 Chern Insulator

### 2.1 Coupled wire picture

The sequential adiabatic evolution for generating the Chern insulator state can be understood intuitively within continuum field theory, using the following scheme based on the coupled wire construction [12,13]. We illustrate the scheme for four wires as depicted in Fig. 2.

The starting point of the evolution is an atomic insulator, which can be thought of as an array of decoupled quantum wires. We describe each wire in terms of a continuum effective theory with a pair of left and right moving fields that are gapped out by scattering between them (see Fig. 2a). Denoting the left and right moving fields by $\psi_{Li}$ and $\psi_{Ri}$, with $i = 1, \dots, 4$ the wire index, the initial continuum Hamiltonian is $H_0 = \sum_i H_{0,i}$, where

$$H_{0,i} = \int dx \left( \psi_{Li}^\dagger \partial_x \psi_{Li} - \psi_{Ri}^\dagger \partial_x \psi_{Ri} + m \psi_{Li}^\dagger \psi_{Ri} + m \psi_{Ri}^\dagger \psi_{Li} \right), \tag{1}$$

where $x$ is the coordinate along the wires.

For the first step of the sequential evolution, we focus on wires 1 and 2 and define the four-component field

$$\Psi = \begin{pmatrix} \psi_{R1} \\ \psi_{L1} \\ \psi_{R2} \\ \psi_{L2} \end{pmatrix}. \tag{2}$$

We introduce $4 \times 4$ matrices $\tau^a = \sigma^a \otimes \mathbb{1}$ and $\mu^a = \mathbb{1} \otimes \sigma^a$, where $a = 1, 2, 3$ and $\sigma^a$ are the $2 \times 2$ Pauli matrices. For example,

$$\tau^1 = \begin{pmatrix} 0 & 1 & 0 & 0 \\ 1 & 0 & 0 & 0 \\ 0 & 0 & 0 & 1 \\ 0 & 0 & 1 & 0 \end{pmatrix}, \tag{3}$$

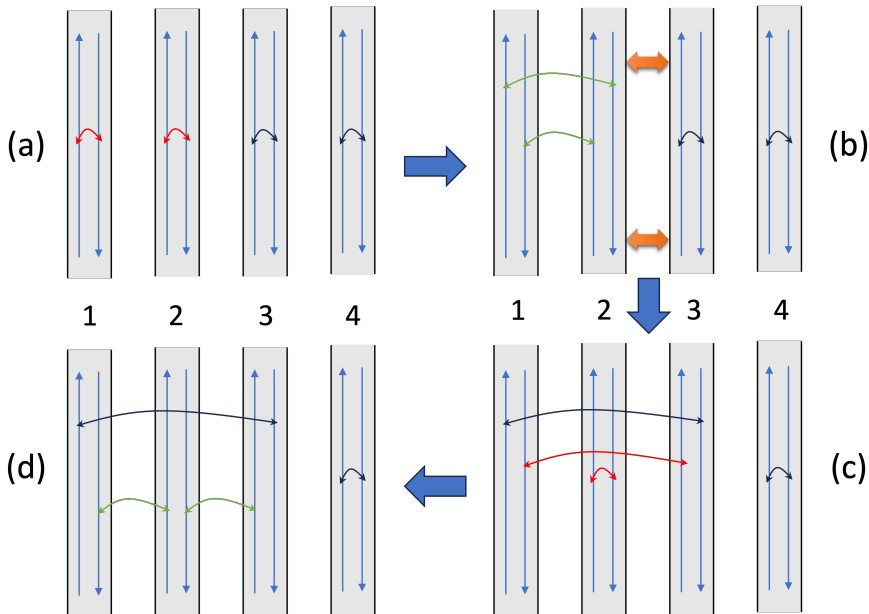

Figure 2: The coupled wire picture of the sequential adiabatic evolution process that generates chiral states. Starting from gapped decoupled wires (a), 1. first tune the intra-wire coupling in wires 1 and 2 (red in (a)), then turn down this coupling while turning up inter-wire coupling between the two wires (green in (b)); 2. exchange wire 2 and 3; 3. first tune the couplings shown in red in (c), then turn down these couplings while increasing those shown in green in (d). Upward (downward) arrows indicate right-moving (left-moving) modes. See the text for more details.

and

$$
\mu^1 = \begin{pmatrix} 0 & 0 & 1 & 0 \\ 0 & 0 & 0 & 1 \\ 1 & 0 & 0 & 0 \\ 0 & 1 & 0 & 0 \end{pmatrix}.
\tag{4}
$$

In this notation, the initial Hamiltonian density for wires 1 and 2 is

$$
\mathcal{H}_0 = \Psi^\dagger(-\tau^3\partial_x + m\tau^1)\Psi.
\tag{5}
$$

The first step of the sequential evolution begins by rotating the mass term from $m\tau^1 \to m\tau^2$, via the Hamiltonian density

$$
\mathcal{H}_{1a}(t) = \Psi^\dagger\Big[-\tau^3\partial_x + m\cos\Big(\frac{\pi}{2}\frac{t}{T}\Big)\tau^1 + m\sin\Big(\frac{\pi}{2}\frac{t}{T}\Big)\tau^2\Big]\Psi,
\tag{6}
$$

where the time parameter $t$ varies from $t = 0$ to $t = T$. This form can be obtained from $\mathcal{H}_0$ by the unitary operation $\Psi \to \exp(i\pi t\tau^3/4T)\Psi$, so the energy spectrum remains unchanged and thus gapped for all $t$. Next we choose the time-dependent Hamiltonian

$$
\mathcal{H}_{1b}(t) = \Psi^\dagger\Big[-\tau^3\partial_x + m\cos\Big(\frac{\pi}{2}\frac{t}{T}\Big)\tau^2 + m\sin\Big(\frac{\pi}{2}\frac{t}{T}\Big)\mu^1\tau^1\Big]\Psi,
\tag{7}
$$

where again $t$ ranges from 0 to $T$. The form Eq. (7) is also realized starting from $\mathcal{H}_{1a}(T)$ by a unitary operation $\Psi \to e^{-i\pi t\tau^3\mu^1/4T}\Psi$, so again the spectrum remains gapped. At the end of the first step, wires 1 and 2 are coupled as shown in Fig. 2b, forming a mini version of a Chern insulator, while wires 3 and 4 remain in the atomic insulator state. The Hamiltonian density of wires 1 and 2 is given by

$$
\mathcal{H}_{1b}(T) = \Psi^\dagger(-\tau^3\partial_x + m\tau^1\mu^1)\Psi.
\tag{8}
$$

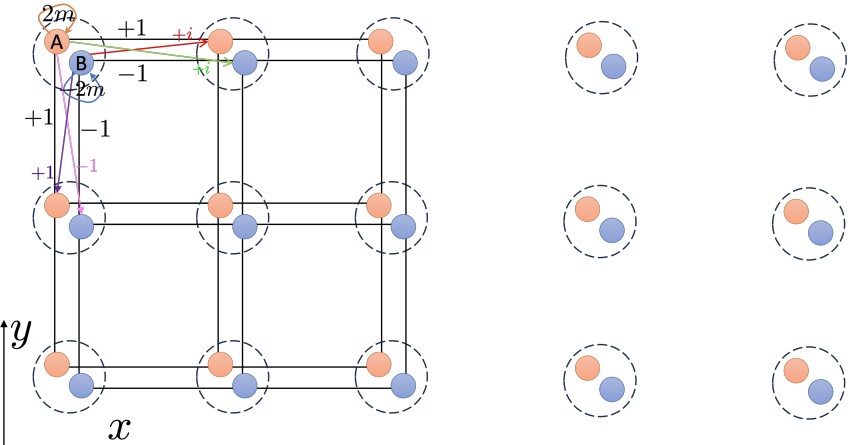

Figure 3: Intermediate step in the Sequential Adiabatic Evolution process for generating the Chern Insulator state from the atomic insulator state. The left half of the system is in the Chern insulator state while the right half of the system is in the atomic insulator state. Labels on arrows indicate coupling strength between neighboring fermion modes. The system is translation invariant in the $y$ direction but not in the $x$ direction.

The next step is to swap the location of wires 2 and 3, giving the system shown in Fig. 2c. Then, in the last step, we adiabatically vary terms coupling wires 1, 2 and 3. However, as we can see from panels (c) and (d) of Fig. 2, the fields $\psi_{R1}$ and $\psi_{L3}$ are not involved in the time-dependent part of the Hamiltonian. Therefore we effectively have another two-wire problem. In terms of the new four-component field

$$\Psi' = \begin{pmatrix} \psi_{R2} \\ \psi_{L2} \\ \psi_{R3} \\ \psi_{L1} \end{pmatrix}, \tag{9}$$

the adiabatic evolution proceeds exactly as in the first step. After this step, wires 1, 2 and 3 form a small Chern insulator. To continue growing the Chern insulator, we move rightward (as depicted in Fig. 2), and repeat the last two steps to add one wire at a time. Once we have incorporated a number of wires proportional to the linear system size, we obtain a 2D Chern insulating state.

Note that throughout the process, the Chern insulator state has periodic boundary conditions, and no gapless chiral edge states are exposed. Below, we describe a similar sequential evolution on the lattice, and study it numerically to verify that the gap remains open. The continuum picture serves as a conceptual guide to the protocol in the lattice model, although we will not be concerned with detailed matching between the lattice Hamiltonian terms and those in the continuum field theory.

## 2.2 Numerical result

In this section, we explicitly construct a gapped adiabatic path to sequentially generate the Chern insulator state from an atomic insulator state. Our construction makes use of the two band model

$$H_c(k_x, k_y) = (m + \cos k_x + \cos k_y)\sigma_z + \sin k_x \sigma_x + \sin k_y \sigma_y. \tag{10}$$

The Chern number is $-1$ if $-2 < m < 0$ and $+1$ if $0 < m < 2$.

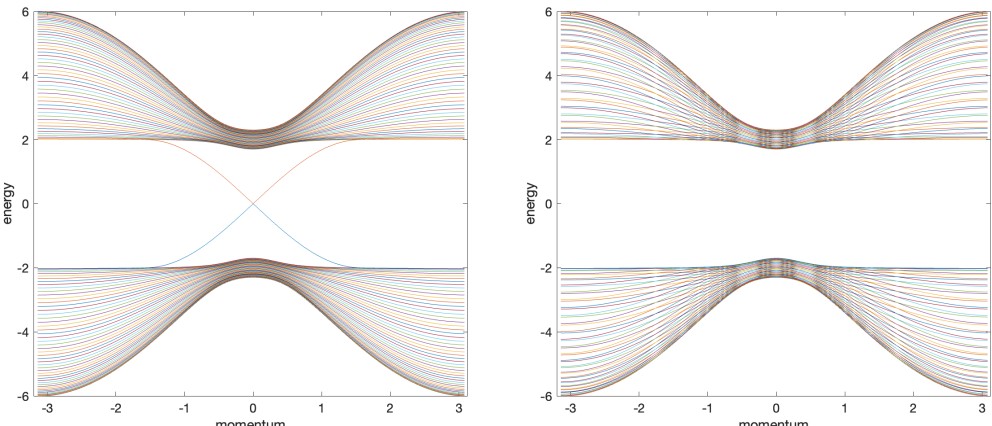

Figure 4: Spectrum of the Chern insulator state with respective to $k_y$ for $N_x = 50$, $m = -1$, $\epsilon = 0.3$. Left: open boundary condition in $x$ direction; Right: closed boundary condition in $x$ direction.

Consider the configuration shown in Fig. 3, where the left half of the system is in the Chern insulator state and the right half of the system is in the atomic insulator state. Suppose that the width of the left half is $N_x$. The system lacks translation invariance in the $x$ direction but has periodic boundary condition in the $y$ direction. The Hamiltonian decouples for each $k_y$ and each $H(k_y)$ block for the Chern Insulator part of the system is of size $2N_x \times 2N_x$. The diagonal $2 \times 2$ block couples the fermion modes within each column $n$ and takes the form

$$H_c(k_y)_n = \begin{pmatrix} c^\dagger_{A,k_y,n} & c^\dagger_{B,k_y,n} \end{pmatrix} \begin{pmatrix} 2m + 2\cos k_y & -2i\sin k_y + \epsilon \\ 2i\sin k_y + \epsilon & -2m - 2\cos k_y \end{pmatrix} \begin{pmatrix} c_{A,k_y,n} \\ c_{B,k_y,n} \end{pmatrix}. \tag{11}$$

An small on-site hopping between $A$ and $B$ of magnitude $\epsilon$ is added to break an accidental symmetry of the system. The coupling terms between $n$ and $n + 1$ are

$$H_c(k_y)_{n,n+1} = \begin{pmatrix} c^\dagger_{A,k_y,n} & c^\dagger_{B,k_y,n} \end{pmatrix} \begin{pmatrix} 1 & i \\ i & -1 \end{pmatrix} \begin{pmatrix} c_{A,k_y,n+1} \\ c_{B,k_y,n+1} \end{pmatrix}. \tag{12}$$

In Fig. 4, we plot the spectrum of $H(k_y)$ vs $k_y$ for $N_x = 50, m = -1, \epsilon = 0.3$. With periodic boundary condition (the $N_x$th column coupled to the 1st column), the spectrum is gapped. With open boundary condition (the $N_x$th column not coupled to the 1st column), the spectrum is gapless with a pair of chiral edge modes.

To add a wire to the Chern insulator half of the system, we first enlarge $H(k_y)$ to size $2(N_x + 1) \times 2(N_x + 1)$ by adding a decoupled $H_c(k_y)_{N_{x+1}}$ block. For $m = -1$, the spectrum of the initially decoupled wire is gapped as long as $\epsilon \neq 0$. Moreover, the wire is in the 1d trivial phase, and can be obtained from an atomic insulator by continuously tuning Hamiltonian terms along the wire. Denoting $H(k_y)$ by $h$, the overall Hamiltonian reads

$$\begin{pmatrix} h_1 & h_{1,2} & & & & h^\dagger_{1,N_x} & \\ h^\dagger_{1,2} & h_2 & & & & & \\ & & \ddots & \ddots & & \ddots & \\ & & & & h_{N_x-1} & h_{N_x-1,N_x} & \\ h_{1,N_x} & & & & h^\dagger_{N_x-1,N_x} & h_{N_x} & \\ & & & & & & h_{N_x+1} \end{pmatrix}. \tag{13}$$

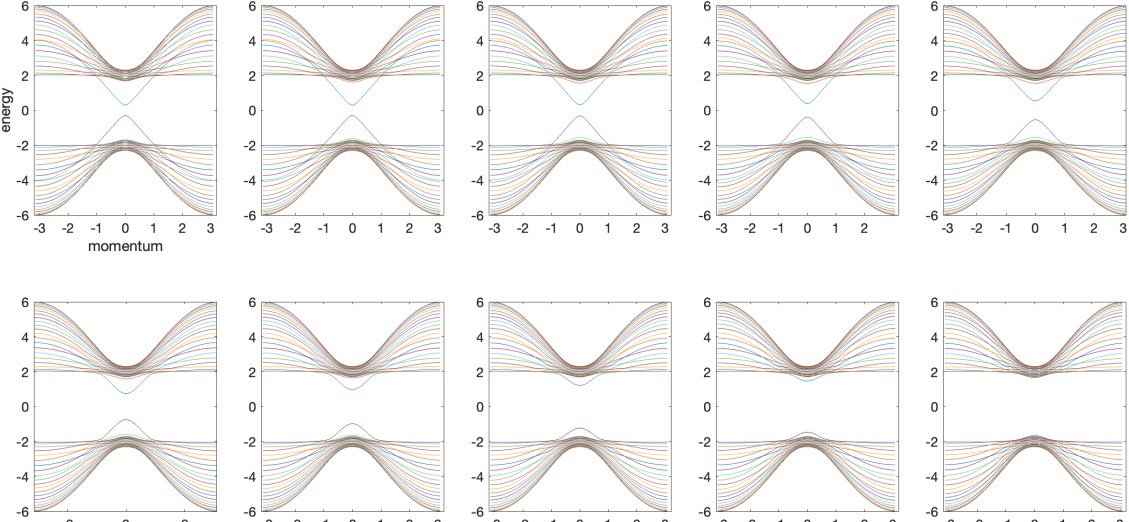

Figure 5: The adiabatic evolution of the spectrum as one extra wire is added to the Chern insulator state with width $N_x = 20$. The spectrum remains gapped as the wire merges into the bulk. In all of the plots, the $x$ axis represents momentum and the $y$ axis represents energy.

The new wire is merged into the Chern insulator state by first swapping column $N_x$ with $N_x+1$, then tuning down the coupling between column $N_x-1$ and $N_x+1$ while tuning up the coupling between column $N_x - 1$ and $N_x$ as well as the coupling between column $N_x$ and $N_x + 1$.

$$
\begin{pmatrix}
h_1 & h_{1,2} & & & & & h^\dagger_{1,N_x+1} \\
h^\dagger_{1,2} & h_2 & & & & & \\
& \ddots & \ddots & & \ddots & & \\
& & & h_{N_x-1} & r(t)h_{N_x-1,N_x} & s(t)h_{N_x-1,N_x+1} \\
& & & r(t)h^\dagger_{N_x-1,N_x} & h_{N_x} & r(t)h_{N_x,N_x+1} \\
h_{1,N_x+1} & & & s(t)h^\dagger_{N_x-1,N_x+1} & r(t)h^\dagger_{N_x,N_x+1} & h_{N_x+1}
\end{pmatrix}.
\tag{14}
$$

We choose the time dependence of the evolution to be $s(t) = 1 - t$, $r(t) = t$. We plot the spectrum of $H(k_y)$ vs $k_y$ as $t$ grows from 0 to 1 in Fig. 5. We see that the spectrum remains gapped during the whole process as the extra wire is added to the Chern insulator, with the spectrum of the wire merging into the bulk bands. In Fig. 5, we plot the spectrum for $N_x = 20$. For other values of $N_x$, the spectrum takes a similar form. The minimal gap in the process remains almost constant when increasing $N_x$. Therefore, progressing wire by wire, we can generate the Chern insulator state from the atomic insulator state with a sequential adiabatic evolution.

## 3 Topological superconductor

A similar sequential adiabatic process can be constructed for the $p + ip$ superconductor. The intuitive picture is the same as that discussed in section 2.1 except that instead of complex fermion left/right movers, we start from Majorana fermion left/right movers. The rest of the picture remains the same.

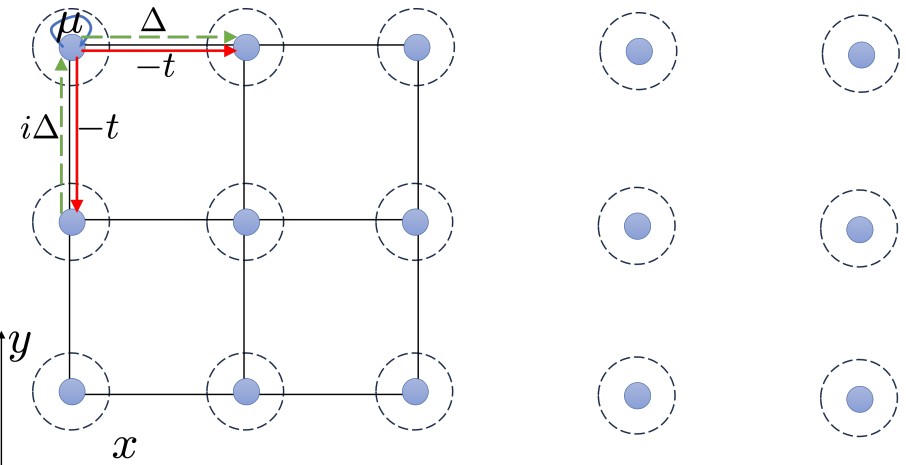

Figure 6: Intermediate step in the Sequential Adiabatic Evolution process for generating the $p + ip$ superconductor from the atomic superconducting state. The left half of the system is in the $p + ip$ state while the right half of the system is in the atomic superconducting state. Labels on solid arrows indicate hopping strength while labels on dashed arrows indicate Cooper pairing strength. The system is translation invariant in the $y$ direction but not in the $x$ direction.

For numerical verification of the existence of a gap in the adiabatic process, we make use of the $p + ip$ Hamiltonian on square lattice

$$H_p = \sum_{\vec{r}} -tc_{\vec{r}}^\dagger c_{\vec{r}+\hat{x}} - tc_{\vec{r}}^\dagger c_{\vec{r}+\hat{y}} + \Delta c_{\vec{r}}^\dagger c_{\vec{r}+\hat{x}}^\dagger + i\Delta c_{\vec{r}}^\dagger c_{\vec{r}+\hat{y}}^\dagger + (\text{h.c.}) - \mu c_{\vec{r}}^\dagger c_{\vec{r}}. \tag{15}$$

Consider the configuration as shown in Fig. 6 where the left half of the system (of width $N_x$) is in the $p + ip$ state and the right half of the system is in the atomic superconductor product state. The system lacks translation invariance in the $x$ direction but has periodic boundary condition in the $y$ direction. The Hamiltonian decouples for each $k_y$ and each $H(k_y)$ block for the $p + ip$ part of the system is of size $2N_x \times 2N_x$. The diagonal $2 \times 2$ block couples the fermion modes within each column $n$ and takes the form

$$H_p(k_y)_n = \begin{pmatrix} c_{n,k_y}^\dagger & c_{n,-k_y} \end{pmatrix} \begin{pmatrix} -t\cos k_y - \mu/2 & -i\Delta e^{-ik_y} \\ i\Delta e^{ik_y} & t\cos k_y + \mu/2 \end{pmatrix} \begin{pmatrix} c_{n,k_y} \\ c_{n,-k_y}^\dagger \end{pmatrix}. \tag{16}$$

The coupling terms between $n$ and $n+1$ are

$$H_p(k_y)_{n,n+1} = \begin{pmatrix} c_{n,k_y}^\dagger & c_{n,-k_y} \end{pmatrix} \begin{pmatrix} -t & -\Delta \\ \Delta & t \end{pmatrix} \begin{pmatrix} c_{n+1,k_y} \\ c_{n+1,-k_y}^\dagger \end{pmatrix}. \tag{17}$$

In Fig. 7, we plot the spectrum of $H(k_y)$ vs $k_y$ for $N_x = 50, \Delta = 1, t = 1, \mu = 2$. With periodic boundary condition (the $N_x$th column coupled to the 1st column), the spectrum is gapped. With open boundary condition (the $N_x$th column not coupled to the 1st column), the spectrum is gapless with a pair of chiral edge modes.

Now we can follow the same sequential process as described in Eq. 13 and 14 except that $H_c(k_y)$ blocks are replaced by $H_p(k_y)$ blocks. At each step of the sequential evolution, we add a decoupled 1d $p$-wave superconducting wire, in the 1d trivial phase, whose Hamiltonian is given by a decoupled $H_p(k_y)_{N_x+1}$ block. We plot the spectrum of $H(k_y)$ vs $k_y$ as $t$ grows from 0 to 1 in Fig. 8. We see that the spectrum remains gapped during the whole process as the extra wire is added to the $p + ip$ superconducting state (the spectrum of the wire merges into

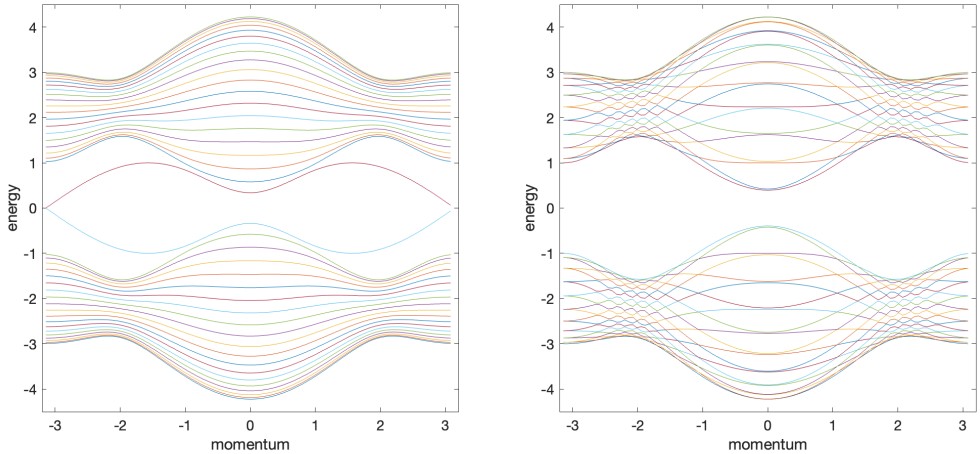

Figure 7: Spectrum of the $p + ip$ superconductor with respective to $k_y$ for $N_x = 50, \Delta = 1, t = 1, \mu = 2$. Left: open boundary condition in $x$ direction; Right: closed boundary condition in $x$ direction.

the bulk bands). In Fig. 8, we plot the spectrum for $N_x = 10$ and 15. For other values of $N_x$, the spectrum takes a similar form. The minimum gap in the process varies a little bit with increasing $N_x$, but no more than 5%. Therefore, progressing wire by wire we can generate the $p+ip$ superconducting state from the atomic superconducting state with a sequential adiabatic evolution.

## 4 Gauging as a sequential circuit

Starting from the free fermion chiral state discussed above, we can demonstrate how to generate strongly interacting chiral states using a sequential unitary transformation. In particular, in this section we show how to couple a state to a finite gauge field using a sequential quantum circuit. Combined with the sequential adiabatic evolution for generating the free fermion chiral states, we can get a sequential unitary transformation for the chiral Ising state, the chiral semion state, or other types of strongly correlated chiral gauge theories.

Consider, without loss of generality, a 2D matter system with $Z_2$ global symmetry. The symmetry operator on each matter degree of freedom (DOF) is given by an unitary operator $X$. The global symmetry is given by $\prod_i X_i$. In a fermionic system where the $Z_2$ symmetry to be gauged is the fermion parity, the symmetry operator is given by $(-1)^{n_f}$ with $n_f$ the fermion number operator. Replacing $X$ with $(-1)^{n_f}$ in the following discussion gives the gauging circuit coupling a fermion system to a $Z_2$ gauge field.

To couple the system to a $Z_2$ gauge field, we first add $Z_2$ gauge field DOF $\tau$ to the edges of the lattice. Initially, all the $\tau$ DOF are set to be in state $|0\rangle$ stabilized by operator $\tau_z$. The gauge field can be coupled to the matter with the sequential circuit shown in Fig. 9. The first step consists of a finite depth circuit in the row labeled 1. We apply the Hadamard gate to the orange gauge DOFs, then apply the controlled-Not gates indicated by the black arrows. The gate set within each dotted red box maps a single $\tau_z$ operator acting on the orange gauge DOF to $X_i \prod_{i \in e} \tau_x^e$, the Gauss' law term in the gauge theory. The gate sets in all the red dotted boxes in the first row commute with each other and can be applied at the same time. This finite depth circuit can be applied row by row (step 1 to 2) until we reach the last row where the gate sets must be applied box by box and rotated (step 3, 4, 5), upon which the circuit is

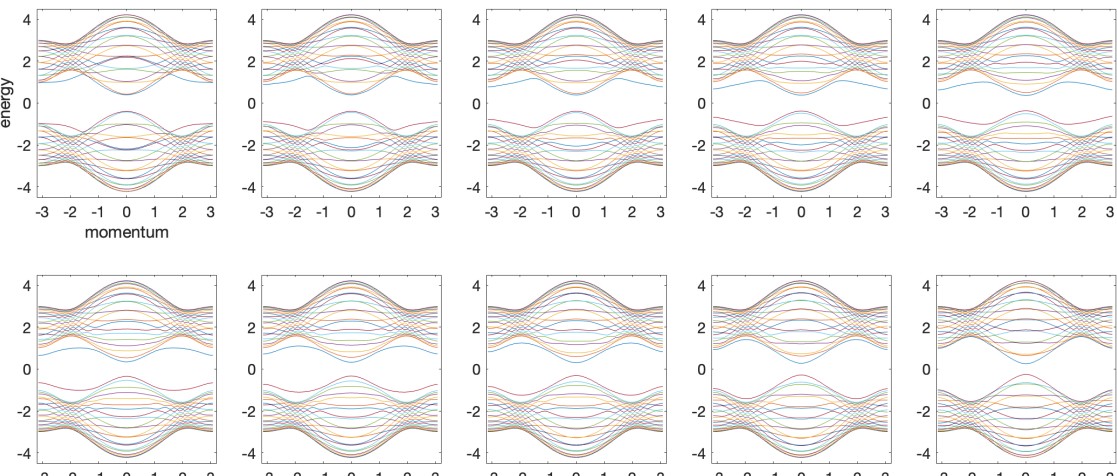

Figure 8: The adiabatic evolution of the spectrum as one extra wire is added to the $p + ip$ superconducting state with width $N_x = 20$. The spectrum remains gapped as the wire merges into the bulk.

complete. The state is now stabilized by all the Gauss' law terms at each vertex. Moreover, we can check that the minimal coupling terms $Z_i Z_j$ are now dressed by a string of $\tau_z$'s in between so that they are gauge invariant. The gauge DOFs which do not turn into Gauss' law terms become flux terms of the form $\prod_{e \in p} \tau_z^e$ around plaquettes $p$. Therefore, after the circuit, the matter DOF are coupled to the $Z_2$ gauge field.

This circuit is very similar in form to the one discussed in [1, 15] to generate the toric code state from the product state. The only difference is that the Toric Code discussed in [1, 15] is a pure gauge theory (within the low-energy subspace of the Gauss' law term) while here we include the matter DOF as well. Explicitly, if the matter DOFs are originally in the trivial paramagnetic state $|++...+\rangle$, then the final gauge-matter state is exactly the ground state of the $Z_2$ lattice gauge theory.

Replacing the operator $X$ in the circuit shown in Fig. 9 with the fermion parity operator $(-1)^{n_f}$ gives the fermionic gauging circuit that couples a fermionic state to the $Z_2$ gauge field. In particular, in the contolled-Not gate (controlled by a $\tau_x$), the $X$ operator is replaced with the fermionic parity operator $(-1)^{n_f}$. The fermion parity operator is fermion even. Therefore, the replacement does not lead to any issue of non-commutativity among operators.

## 5  Sequential adiabatic evolution versus sequential circuits

Some comments are in order about the locality of the sequential adiabatic evolution process. Consider generating the chiral state on a flat 2D plane. This issue can be discussed either in terms of tuning Hamiltonian couplings, or in terms of the locality properties of a unitary mapping that we apply to the initial product ground state.

We first discuss the Hamiltonian tuning picture. At all points in the process we have short-range couplings between neighbouring wires and long-range couplings between the left and right edges of the chiral region. Growing the chiral region requires two steps: swapping a pair of wires (Figs. 2(b)-2(c)) and then tuning the couplings near wire $N_x$ to incorporate the new wire into the chiral region (Figs. 2(c)-2(d)). Within the Hamiltonian picture, the first step requires tuning the long-range couplings, even though the corresponding unitary gates are local. However, the second step only requires tuning of short-range couplings.

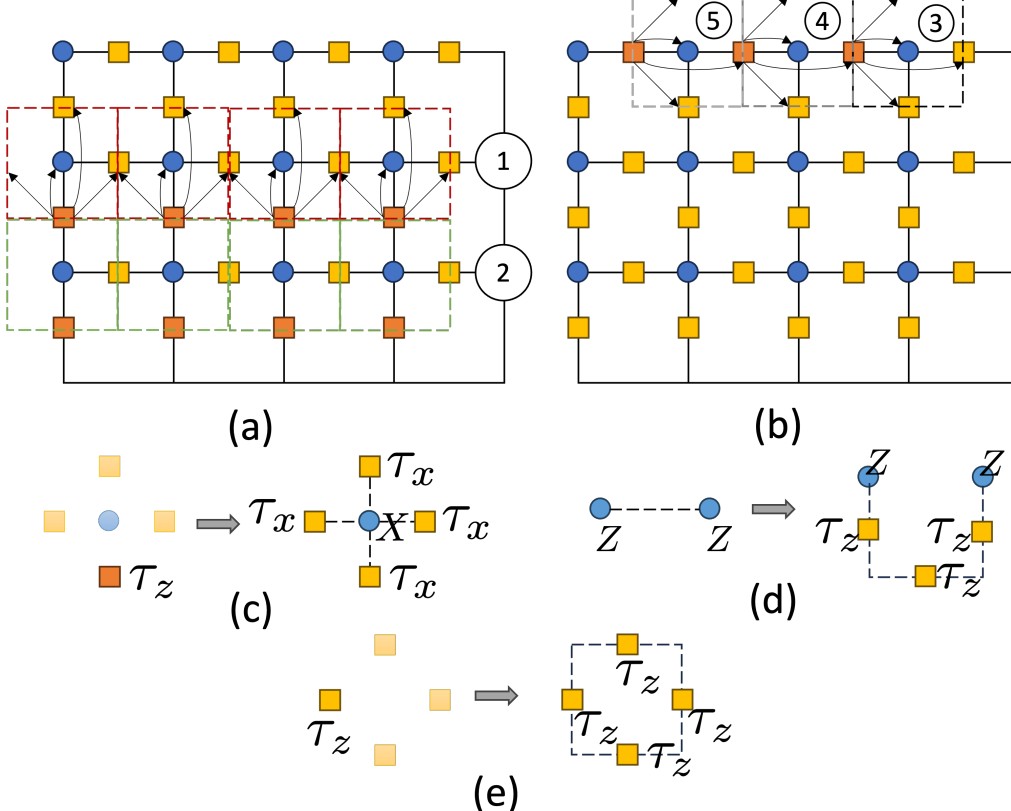

Figure 9: The sequential circuit for coupling matter DOFs (blue dots) with $Z_2$ gauge field DOFs (yellow squares). (a) each dotted box contains a set of Controlled-Not gates represented by the black arrows. In the first/second step of the sequential circuit, the gate sets in the first/second row are applied at the same time. (b) In the third, fourth, fifth step of the sequential circuit, gate sets in box 3,4,5 are applied. After the circuit, (c) $\tau_z$ on orange gauge field DOFs are transformed into Gauss' law terms around each matter DOF; (d) minimal coupling terms $Z \otimes Z$ of the matter DOFs are connected by Wilson line of the gauge field in between; (e) $\tau_z$ on the yellow gauge field DOFs are transformed into gauge flux terms around each plaquette.

We can convert the sequential adiabatic evolution into a unitary mapping between initial (trivial) and final (chiral) ground states. First, swapping the two wires can simply be realized by a parallel series of unitary swap gates between the degrees of freedom in the two wires. Next, we modify Hamiltonian terms near wire $N_x$. Using the quasi-adiabatic continuation [16], we can convert this adiabatic path into a unitary evolution of the ground state. Since we only change Hamiltonian terms near wire $N_x$, and the ground state has a finite correlation length, we expect that the strength of this unitary evolution decays exponentially away from wire $N_x$. However, due to the periodic boundary conditions of the chiral region in the $x$ direction, which are enforced at all times, wire $N_x$ is entangled with wire 1. Therefore, the tails of adiabatic time evolution wrap around the periodic boundary conditions, and extend from the left edge of the lattice, as shown in Fig. 10. This means that the unitary operator needed to grow the chiral region by one step is non-local as it couples the left and right edges. In the $y$ direction, the unitary evolution will be translationally invariant and locally generated since the Hamiltonian terms remain local in the $y$-direction.

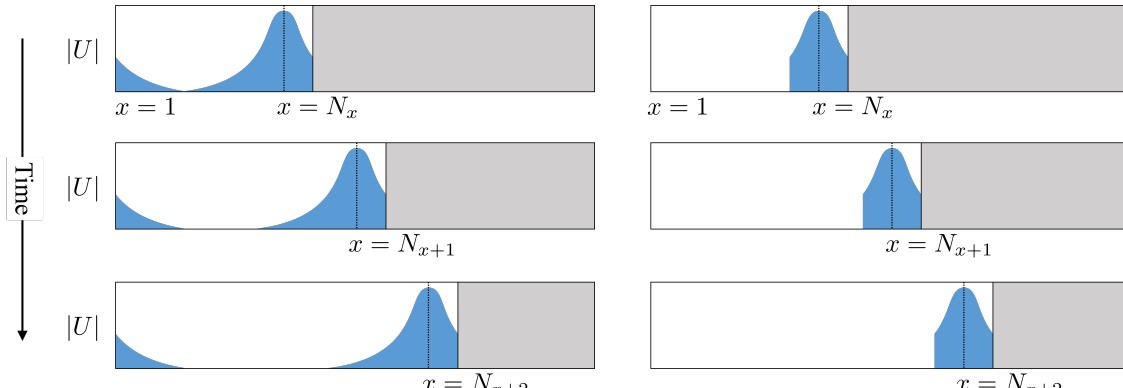

Figure 10: The effective unitary evolution of the ground state during the sequential adiabatic evolution as the chiral region (light) is extended into the trivial region (dark). Left: visualization of the exponentially decaying support of the unitary evolution operator $U$ as a function of the $x$ coordinate (the strength is independent of the $y$ coordinate). Notice that each step involves a non-local unitary that acts on both the left and right edges of the chiral region. Right: If we cut the tails of the unitaries, then each unitary acts only on the right edge of the chiral region and we recover a sequential circuit of local unitaries. The unitary at each step is a finite-depth circuit along the $y$-direction.

To summarize, in both the Hamiltonian tuning and unitary evolution pictures, the sequential adiabatic evolution process has some degree of non-locality. It is interesting to remark that the non-local step in one picture corresponds to a local step in the other picture. We can compare this situation to the sequential circuits used to generate symmetry-protected topological states from Ref. [1]. If we were to convert these circuits into paths of gapped Hamiltonians, we would also find that non-local terms coupling the left and right edges are necessary to preserve the gap, and these non-local terms are tuned during the sequential evolution. However, in this case, in the unitary evolution picture, the unitary gates at each step are strictly local with no tails. This is possible due to the nature of the SPT fixed-point wavefunctions which have zero correlation length. In contrast, the chiral models we considered here necessarily have exponentially decaying correlations.

Within the unitary evolution picture, a local evolution can be obtained by first using swap gates to place the new wire to be incorporated a fixed finite distance away from the boundary of the chiral region. Then, the tails of the unitary obtained from adiabatic evolution can be truncated beyond the same distance in the $x$-direction. However, it is not clear if this can be done without affecting the chiral physics we wish to capture. If so, the above adiabatic evolution takes the form of a sequential quantum circuit, consisting of finite depth circuits applied between columns sequentially from left to right, see Fig. 10. We remark that our initial state was an atomic insulator state, i.e. a product state. At first glance, it may seem like such a sequential circuit cannot generate the necessary exponentially decaying correlations in both directions. However, it turns out they can, as was recently observed in Ref. [17].

Note that exponentially decaying correlation alone does not forbid the possibility of sequential circuit generation. Indeed, it is known that matrix product states can all be generated using sequential circuits, even if they have nonzero correlation length [2, 3]. Another possible obstruction to removing the tails in the chiral state generating process comes from the fact that it would lead to a state which can be represented by a tensor network with finite bond dimension. It is often stated that such tensor networks states cannot capture chiral states with exponentially decaying correlations in the bulk. This can be proven for free-fermionic ten-

sor networks [18], but recent numerical investigations suggest that they can at least provide faithful approximations [19]. Therefore, the extent to which we can truncate the tails in our unitary evolution while preserving the chiral physics remains an interesting open question.

# 6 Discussion

The chiral states covered by methods presented in this paper are either free fermion states or gauge theories with a finite gauge group. The method discussed does not directly apply to states like the chiral Fibonacci state. We expect that a similar sequential adiabatic evolution process for such states can be established, although to demonstrate the validity of the process we need to perform strongly correlated numerical simulation.

The sequential quantum circuit for coupling to a finite gauge field works for any symmetric state, gapped, gapless, or even symmetry breaking. It also applies to more general forms of symmetries like higher form symmetry, sub-system symmetry, etc. Therefore, this kind of circuit should be useful for further investigations of the gauging of many-body systems, including its implementation on quantum devices.

# Acknowledgments

We are grateful for inspiring discussions with Michael Levin and Carolyn Zhang.

**Funding Information** This work is supported by the Simons Collaboration on Ultra-Quantum Matter, funded by grants from the Simons Foundation (651438, XC; 651440, MH, DTS). XC is also supported by the Simons Investigator Award (Simons Foundation award ID 828078), the Institute for Quantum Information and Matter at Caltech and the Walter Burke Institute for Theoretical Physics at Caltech.

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
