# Peer review of "Sequential Adiabatic Generation of Chiral Topological States"

_SciPost Physics, doi:SciPost Phys. 19, 107 (2025)_

## Round 1 · Referee Report · Anonymous (Referee 1) · 2024-10-28

Strengths

1- The manuscript contains a plausible strategy to prepare chiral topological states, a family that previous works could not address. 2- The claim is supported by a numerical simulation of a free fermion model, which convincingly establishes that there exist some models for which this strategy succeeds. 3- The manuscript is clearly written. 4- The connection to sequential circuits is discussed in Sec 5 and offers a number of interesting open problems.

Weaknesses

1- Since there exist efficient algorithms to compile time evolution into quantum circuits, it is not clear why adiabatic Hamiltonian evolution is fundamentally different from quantum circuits. 2- No comparison is attempted with other existing approaches to prepare free-fermion states. 3- A gap is demonstrated numerically for two different system sizes and it's claimed that it remains constant, but it is not explained why this should be the case. 4- Beyond the Hamiltonian path for the topological insulator, the discussion is rather high level and sketched.

Report

The manuscript is insightful and a creative point of view on an important problem. As such it fulfils the SciPost acceptance criteria, especially #2: Open a new pathway in an existing or a new research direction, with clear potential for multi-pronged follow-up work.

I recommend acceptance, provided some of the manuscript shortcomings are addressed.

Requested changes

1- The overall goal of the paper seems to be to provide an efficient way of preparing chiral topological states. In my opinion, the Introduction should briefly mention other approaches one could take, and discuss their shortcomings. If there are no competing approaches to prepare chiral states, this could also be mentioned.

2- In the discussion on relation to quantum circuits, long-range/non-local interactions are mentioned. I think a bit of clarity could be added here by explaining that they arise merely due to moving from a torus to a flat 2D lattice.

3- I don't understand the discussion around exponential tails potentially being an obstruction to preparation through circuits. Already from the MPS example it's clear that a sequential circuit can prepare states with any sort of exponential decay. Of course it's probably not true that a sequential circuit can produce any sort of correlations as long as they are exponentially decaying, but this is also not clear in sequential adiabatic preparation.

Recommendation

Publish (meets expectations and criteria for this Journal)

  • validity: high
  • significance: high
  • originality: high
  • clarity: high
  • formatting: good
  • grammar: perfect

Author:  David Stephen  on 2025-07-25  [id 5681]

(in reply to Report 1 on 2024-10-28)
Category:
answer to question
pointer to related literature

We thank the referee for carefully reading our paper and for the helpful suggestions.

--- The overall goal of the paper seems to be to provide an efficient way of preparing chiral topological states. In my opinion, the Introduction should briefly mention other approaches one could take, and discuss their shortcomings. If there are no competing approaches to prepare chiral states, this could also be mentioned.

The goal of this paper is not quite to provide an "efficient" way of preparing the chiral states. We want to answer the conceptual question of whether a sequential procedure can be used to generate the chiral state that's conceptually similar to the sequential circuit we used to generate non-chiral states. Special emphasis is put on ensuring the unitarity and locality of the protocol. If one were to focus on the efficiency of the protocol, one might consider using non-local gates or non-unitary transformations.

The only other protocol we are aware of is arxiv 2304.13748, which uses an RG like scheme to generate chiral states. As the length scale gets larger and larger in the RG protocol, the evolution involves nonlocal coupling terms. Because of this, we don't consider this other protocol as a competing protocol. But we did cite this paper in the introduction. We added a comment to make it clear that because of the involvement of the adiabatic evolution of nonlocal coupling terms, this protocol has a different setup than the one we are considering.

--- In the discussion on relation to quantum circuits, long-range/non-local interactions are mentioned. I think a bit of clarity could be added here by explaining that they arise merely due to moving from a torus to a flat 2D lattice.

We can think of three different potential generalizations to the torus (the cross section of which is illustrated in the attached figure), but none of them solves the nonlocality issues, each for different reasons.

In (a), the torus grows as the state is sequentially generated. The torus has to be embedded in 3d space with a uniform density of qubits per volume as a resource. The circuit has to act on the whole 2d torus at each sequential step (to move it outward at each step), moreover, the circuit winds up living in 3d instead of in 2d. So this becomes rather far from what we might mean by a 2d sequential circuit or sequential adiabatic evolution.

If we are only allowed to grow the torus at one location while keeping other parts intact, the geometry becomes (b). This procedure can be carried out strictly in 2d (in a bilayer 2d system), but one essentially gets a “folded” chiral state, because the top and bottom halves of the torus are close together. Therefore, we should think of this state as a stack of two states with opposite chiralities, and not a chiral state.

Finally, in (c), the state partially covers a large torus when it grows (indicated by the red arrow). We encounter the same nonlocality issue as on the 2d plane when the size of the state becomes a finite fraction of the torus.
We have added a comment at the beginning of section 5 to make it clear that we are considering a flat 2D plane geometry.

--- I don't understand the discussion around exponential tails potentially being an obstruction to preparation through circuits. Already from the MPS example it's clear that a sequential circuit can prepare states with any sort of exponential decay. Of course it's probably not true that a sequential circuit can produce any sort of correlations as long as they are exponentially decaying, but this is also not clear in sequential adiabatic preparation.

It is true that MPS with a finite correlation length can be generated with sequential circuits, so the finite correlation length of chiral states is not by itself a reason why a sequential circuit generation is not possible. We suspect that a sequential circuit generation would be impossible for chiral states because otherwise there would be a finite bond dimension tensor network representation of chiral states, which is generally believed not to be true. In any case, all we are saying here is that if we naively convert the adiabatic process into a circuit, it does not give a circuit with strictly local gates. We added a comment in the last paragraph that "exponentially decaying correlation alone does not forbid the possibility of sequential circuit generation. Indeed, it is known that matrix product states can all be generated using sequential circuits, even if they have nonzero correlation length."

Attachment:

---

## Round 1 · Referee Report · Anonymous (Referee 2) · 2024-11-11

Report

This manuscript generalizes the previous unitary sequential circuits to the case of chiral phases of matter. The focus is on free-fermion states, which is motivated based on field-theoretic approaches and confirmed via lattice simulations. The authors also comment on how this can apply to interacting chiral states via a gauging procedure. Finally the authors point out subtleties related to exponential tails and truncations.

This is an interesting work and while the construction is natural and convincing, I would not have expected that one can prepare chiral states in this way (before seeing the solution). The idea of growing a system whilst preserving periodic boundary conditions is rather nice. It shows how chiral states fit into the idea of sequential unitary circuits (or Hamiltonian generalizations thereof), and it also opens some interesting follow-up directions related to the truncated versions of such circuits. This work thus manages to make some surprising connections to recent concepts of interest, and motivates potential follow-up works. In principle I would like to recommend publication, but there is one part of the paper where I am not yet fully convinced:

In Sec 4, the authors discuss gauging as a sequential circuit. This seems correct for the case that is written out, namely a bosonic Z_2 symmetry. But the authors want to use it for the case where Z_2 is fermion parity. There might be additional subtleties in this case, similarly to how the 2+1d fermionization map is subtle. Since this part of the paper is essential to making the case that one can build e.g. chiral Ising anyon theories, I can only recommend publication if the authors include the circuit for the case where the symmetry is fermion parity. Note that this requires some changes, since the controlled-Not gates would naively become fermionic gates which sounds problematic and requires additional care.

In addition, I had a few smaller points for the authors' consideration:

  • Fig 5 has no labels

  • Sec 4 (gauging as circuit): is it interesting to also add the extra FDLU step which disentangles "X prod tau^z = 1" --> "X = 1", so one is left over with just the gauge field?

  • Sec 4: "This circuit is very similar in form to the one discussed in [1] to generate the toric code state from the product state" -> should perhaps also cite https://arxiv.org/abs/2104.01180 ?

  • Sec 4 feels rather brief / ends abruptly. Worth adding an explicit example?

  • Sec 6: typo "Fibbonnaci state"

  • Sec 6: section ends very abruptly (especially with "etc")

  • The abstract mentions "despite the fact that they lack an exactly solvable form" but that is not quite right since of course the main examples are exactly solvable (free-fermion)

Recommendation

Ask for minor revision

  • validity: good
  • significance: good
  • originality: high
  • clarity: good
  • formatting: reasonable
  • grammar: good

Author:  David Stephen  on 2025-07-25  [id 5682]

(in reply to Report 2 on 2024-11-11)
Category:
answer to question

We thank the referee for carefully reading our paper and for the helpful suggestions.

--- In Sec 4, the authors discuss gauging as a sequential circuit. This seems correct for the case that is written out, namely a bosonic Z_2 symmetry. But the authors want to use it for the case where Z_2 is fermion parity. There might be additional subtleties in this case, similarly to how the 2+1d fermionization map is subtle. Since this part of the paper is essential to making the case that one can build e.g. chiral Ising anyon theories, I can only recommend publication if the authors include the circuit for the case where the symmetry is fermion parity. Note that this requires some changes, since the controlled-Not gates would naively become fermionic gates which sounds problematic and requires additional care.

The generalization of the circuit in Fig. 9 to the fermionic case is straightforward. One just need to replace the $X$ in the controlled-Not operator (controlled by a $\tau_x$) with the fermionic parity operator $(-1)^{n_f}$. This controlled-$(-1)^{n_f}$ gate is a well-defined local, fermionic even, unitary gate acting in the gauge-matter Hilbert space. No issues with fermionization arise since we are dealing with physical fermionic degrees of freedom. Therefore, the replacement does not lead to any issue of non-commutativity among operators. We have added a paragraph at the end of section 4 to explain this point.

--- Fig 5 has no labels

For all the subfigures in Fig. 5, the y axis represents energy while the x axis represents momentum. To avoid redundancy in the plot, we put labels only in the first subfigure. We have added an explanation about this in the caption of Fig. 5.

--- Sec 4 (gauging as circuit): is it interesting to also add the extra FDLU step which disentangles "X prod tau^z = 1" --> "X = 1", so one is left over with just the gauge field?

One can add such a step. It simplifies the model but the topological order of the model remains the same. Therefore, to address the question we want to address in this paper, we do not think it is necessary to add this step.

--- Sec 4: "This circuit is very similar in form to the one discussed in [1] to generate the toric code state from the product state" -> should perhaps also cite https://arxiv.org/abs/2104.01180 ?

We thank the referee for the suggestion. The reference has been added.

--- Sec 4 feels rather brief / ends abruptly. Worth adding an explicit example?

We thank the referee for the suggestion. We added a paragraph now about generalizing the circuit to fermionic matter degrees of freedom. We also added some discussion of examples near the end of section 4.

--- Sec 6: typo "Fibbonnaci state"

Thanks, corrected.

--- Sec 6: section ends very abruptly (especially with "etc")

We have added a sentence "Therefore, this kind of circuit should be useful for further investigations of the gauging of many-body systems, including its implementation on quantum devices."

--- The abstract mentions "despite the fact that they lack an exactly solvable form" but that is not quite right since of course the main examples are exactly solvable (free-fermion)

Thanks for pointing this out. We made it more clear that we meant "a zero-correlation-length" exactly solvable form.

---

## Round 2 · Author Response

In response to referee comments, we have made several clarifications throughout the manuscript, as listed below discussed in more detail separately in each reply.

---

## Round 2 · List of Changes

• Added comparisons to related protocols for preparing chiral topological states

  • Clarified the gauging sequential circuit for the case of fermionic parity

  • Clarified the roles of locality and exponentially decaying correlations

---

## Editorial Decision

published